# The Right Temporal Lobe and the Enhancement of Voice Recognition in Congenitally Blind Subjects

**DOI:** 10.3390/brainsci13030431

**Published:** 2023-03-02

**Authors:** Stefano Terruzzi, Costanza Papagno, Guido Gainotti

**Affiliations:** 1Centro di Riabilitazione Neurocognitiva, CIMeC (Center for Mind/Brain Sciences), University of Trento, 38068 Rovereto, Italy; 2CISMed (Centro Interdipartimentale di Scienze Mediche), University of Trento, 38122 Trento, Italy; 3Memory Clinic, Department of Science of Elderly, Neuroscience, Head and Neck and Orthopaedics, Fondazione Policlinico A. Gemelli, IRCCS, 00136 Rome, Italy

**Keywords:** blind people, voice recognition, right temporal lesions

## Abstract

Background: Experimental investigations and clinical observations have shown that not only faces but also voices are predominantly processed by the right hemisphere. Moreover, right brain-damaged patients show more difficulties with voice than with face recognition. Finally, healthy subjects undergoing right temporal anodal stimulation improve their voice but not their face recognition. This asymmetry between face and voice recognition in the right hemisphere could be due to the greater complexity of voice processing. Methods: To further investigate this issue, we tested voice and name recognition in twelve congenitally blind people. Results: The results showed a complete overlap between the components of voice recognition impaired in patients with right temporal damage and those improved in congenitally blind people. Congenitally blind subjects, indeed, scored significantly better than control sighted individuals in voice discrimination and produced fewer false alarms on familiarity judgement of famous voices, corresponding to tests selectively impaired in patients with right temporal lesions. Conclusions: We suggest that task difficulty is a factor that impacts on the degree of its lateralization.

## 1. Introduction

A prevalent involvement of the right hemisphere in processes underlying identity voice recognition has been revealed by experimental investigations and clinical observations which have shown that not only face but also voice processing is predominantly underpinned by the right hemisphere. A right lateralization of the temporal voice areas (TVAs) bilaterally located in the superior temporal sulcus and gyrus [1,2,3] has been, in fact, demonstrated by von Kriegstein et al. [4,5] who have documented a more important role of these structures in voice (compared with speech) recognition. These experimental data have been accompanied by clinical observations (e.g., [6,7,8,9,10,11,12,13,14,15]) which have shown that not only face but also voice processing is predominantly underpinned by the right hemisphere. Pursuing this line of clinical investigations, Papagno et al. [16] tried to compare results obtained by two groups of patients with either a right or a left temporal glioma on various aspects of voice processing (unknown voice discrimination, familiarity assessment, false alarms, personal semantic retrieval, and name retrieval from famous voices judged as familiar). These authors used for their study the ‘Famous People Recognition Battery’ [13], in which subjects are requested to discriminate unknown faces and voices and recognize persons well-known at the Italian national level by evaluating familiarity and identification processes through their faces and voices. The results obtained on these tasks documented that patients with right-sided gliomas were significantly more impaired in voice discrimination and produced more false alarms (FAs) on voice familiarity evaluation than those with left-sided gliomas. The interest of these observations was increased by the results obtained in clinical [17] and experimental [18] investigations, which have shown that in patients with damage to the right anterior temporal lobe (ATL), this pattern of selective impairment (i.e., selective defects of perceptual discrimination and the number of FAs) was higher for voice than for face recognition. These differences between face and voice recognition modalities could be due either to the location of brain damage in these patients or to a greater difficulty met in person recognition through the voice in comparison to the face. The first interpretation stems from the fact that damage to temporal cortices could mainly affect voice processing, whereas face processing could be more clearly affected by lesions involving the inferior temporo-occipital cortices. On the other hand, the second interpretation is supported by the observation that asymmetries between these recognition modalities have been documented in the field of experimental social psychology by several authors (e.g., [19,20,21,22,23]) who have shown that (a) it is more difficult to recognize a person through his/her voice than through his/her face and (b) FAs are produced more in voice than in face recognition. In a further study, Papagno et al. [24] tried, therefore, to investigate more in depth all these issues using again the ‘Famous People Recognition Battery’ [13] to assess FAs during recognition of famous people from faces and voices in patients with right and left ATL tumours and in normal participants tested after anodal transcranial direct current stimulation (tCDS) over the left or right ATL. In patients with unilateral temporal tumours, lesion side did not differentially affect patients’ sensitivity to discriminate or response criterion to recognize famous faces. On the contrary, on the voice recognition task, a lower sensitivity index and lower response criterion were found in patients with right temporal tumours than in those with left-sided lesions. Furthermore, greater right-sided involvement in voice than in face processing was confirmed by the observation that in normal subjects, right ATL anodal stimulation significantly increased voice but only marginally influenced face sensitivity. Taken together, these results suggested that the asymmetry between face and voice recognition in the right hemisphere could be due to the greater complexity of voice processing and to the difficulty of forming stable and well-structured representations, which are necessary to evaluate if a presented voice matches or not with a nonexisting voice representation, producing a higher rate of FA.

Since the meaning of these hemispheric asymmetries remains highly hypothetical, we wondered whether it could be at least in part clarified by studying identity voice recognition in a condition very different from brain damage or brain stimulation but also capable of influencing functional brain organization and hemispheric asymmetries, namely, in congenitally blind subjects. It is, indeed, frequently reported that blind people perform better than sighted ones on a variety of nonvisual tasks, and, in particular, on voice processing tasks (e.g., [25,26,27,28]), probably because they compensate for their lack of vision with increased processing within other sensory modalities, i.e., recruiting occipital cortices during voice processing. Specific information about the pathways through which auditory information allowing voice recognition might reach the visual cortex in congenitally blind subjects have been obtained by human studies which have provided indirect evidence for the existence of axonal connections between face processing areas in the fusiform gyrus and voice processing areas in the superior temporal sulcus (STS) [29,30,31]. The existence of these direct connections between face processing areas in the fusiform gyrus and voice processing areas in the STS has led many authors to assume that congenital visual deprivation may induce an expansion of these connections, which could lead to a reallocation of voice identity processing from the STS to the fusiform gyrus. Drawing on this hypothesis, with functional magnetic resonance imaging (fMRI) studies, Holig et al. [28] and Dormal et al. [32] investigated the changes in the functional organization of neural systems that could be involved in voice identity processing in congenital blindness. In the Holig et al.’s study [28], the brain systems mediating voice identity processing were assessed with a priming paradigm in which two (personally congruent vs. incongruent) voice stimuli were subsequently presented. Person-incongruent compared with person-congruent voices elicited an increased activation in the right anterior fusiform gyrus in congenitally blind individuals but not in matched sighted control participants, whereas the same contrast elicited a higher activation in the right posterior superior temporal sulcus in matched sighted controls. In a further fMRI study [33], the same authors obtained similar results in late blind subjects. In fact, in this case too, blind volunteers, but not matched sighted controls, showed an increase of the BOLD signal in the right anterior fusiform gyrus in response to person-incongruent compared with person-congruent trials. Evidence of a selective increase in the functional coupling between the left temporal voice area and the right fusiform gyrus were provided by Dormal et al. [32], studying the functional preference for object sounds and voices in the brain of early blind and sighted individuals. Functional connectivity analyses evidenced, in fact, that with vocal sounds, a selective increase of functional connectivity was obtained in the blind group between the left temporal voice area and the right fusiform gyrus.

All these data not only pointed to a greater role of the right temporal lobe in the development of voice processing abilities of blind subjects but also suggested that the reorganization of brain functions in these subjects may involve aspects of hemispheric asymmetries. Some hints in this direction are also provided by neurophysiological [34] and functional brain imaging investigations (e.g., [35,36]) which have documented a reduced left lateralization of language in congenitally blind individuals. According to Lane et al. [36], this reduced left lateralization of language concerned both the classic fronto-temporal language areas and the recuited “visual” cortices. Furthermore, the haemodynamic responses in the right hemispheric and occipital areas varied as a function of syntactic and semantic processing demands, providing evidence that all these areas are incorporated into the language network.

However, to better evaluate the relations between the impairment of voice recognition documented in patients with right temporal lesions and the improvement observed in congenitally blind individuals, it could be interesting to judge if the same components of voice comprehension are involved in these opposite conditions. We, therefore, thought that this information could be obtained by administering to groups of congenitally blind subjects and of sighted controls the ‘Famous People Recognition Battery’ [13] used by Papagno et al. [16,24] in their previous studies to evaluate if the same components of voice recognition that are impaired in patients with right temporal tumours are improved during voice identity recognition in congenitally blind individuals.

More specifically, it was assumed that if the correlation between voice discrimination errors and the number of FA observed in previous clinical studies [16,24] in patients with lesions of the right temporal lobe was also found in congenitally blind individuals, this could point to a common underlying mechanism. It was also suggested that the presence of this symptom complex in patients with lesions of the right temporal lobe might be due to the complexity of voice processing that could impact on the capacity of this lobe to form stable and well-structured representations, allowing for the evaluation of whether a presented voice matches or not with an already known voice.

## 2. Materials and Methods

Twelve congenitally blind participants (8 F and 4 M) [mean age = 34.33 ± 3.36 (range = 31–42), mean education = 16.25 ± 2.70 years (range = 13–21)] and twelve healthy controls (9 F and 3 M) [mean age = 33.93 ± 2.99 (range = 30–40), mean education = 16.91 ± 2.32 years (range = 13–21)] took part in this study. Demographical and clinical features of blind participants are shown in Table 1. None of the participants had a history of neurological, psychiatric, or neuropsychological problems, and the two groups did not differ in age [t(22) = 0.56, *p* = 0.57, d = 0.23], years of education [t(22) = −0.57, *p* = 0.57, d = −0.23], or Verbal Judgment Test (a test of verbal intelligence [37]) adjusted score [t(22) = −1.26, *p* = 0.22, d = −0.51].

Informed consent was obtained from all subjects involved in this study, which was conducted according to the guidelines of the Declaration of Helsinki and approved by the Ethics Committee.

Both groups were submitted to two tests assessing voice recognition:

**(i) Unknown voice discrimination test (UVD)** [13]. The test consisted of 20 trials. In each trial, two audio files were consecutively presented with a 2 sec delay between them (total time for each trial: about 15 s). Participants were asked to judge whether the two voices that they heard belonged to the same or to different people. The possible interference of the sentence content was reduced by subdividing stimuli into four groups: (1) same voice/same sentence; (2) same voice/different sentence; (3) different voice/same sentence; and (4) different voice/different sentence. In the case of (1), two identical recordings were presented (the sentence was not recorded twice under different conditions but was exactly the same one). For each trial, one point was assigned for each correct response, and the total score (range: 0–20) was then adjusted for age and education level (“adjusted score”) according to the parameters estimated in a normal sample (193 neurologically unimpaired subjects) with a multiple regression model [13].

**(ii) Familiarity judgment, recognition, and naming of famous people from voices (VO-REC)** [13]. The test included 60 items: 40 voices of very well-known people nationwide (famous voices) and 20 nonfamous voices. Participants were asked to listen to audio fragments and provide a familiarity judgment (i.e., “is this voice familiar to you?”). Each audio fragment lasted about 15 s and did not contain any element that could allow the direct recognition of the person. One point was assigned for each correct response (total score: 0–60). If the familiarity judgment was positive for famous people’s voices, participants were asked three further questions (semantic score). The first two questions were multiple choice questions and investigated the general and specific categories to which the famous person belonged: 1) example of general information (first question): “is this person involved in: (a) politics; (b) entertainment; (c) sport; (d) society?” (2) Example of specific information (second question): “is this entertainer involved in: (a) cinema; (b) theatre; (c) music; (d) TV?” The third question was instead an open question asking participants to provide unequivocally identifying information about the person (i.e., movies titles, political role, and party, and so on). One point was assigned for each correct response (total score: 0–120). Finally, for voices judged as familiar, participants were asked to provide the name of that person. Naming score was computed as a percentage of voice assessed as familiar [38]. Furthermore, a false alarms score (FAs) was calculated: This corresponded to the number of nonfamous voices judged as familiar (total score: 0–20). Total scores for familiarity, semantic, naming, and false alarms have been adjusted for age and education level (“adjusted score”) according to the Italian normative data [13,38].

In addition to the previous two tests, a slightly modified version of the familiarity evaluation and person identification from name (NA-REC) [39] was administered too. In the original version of the test, written names of the same 40 Italian celebrities whose voices had been presented in the VO-REC are used. Participants are asked to recognize these famous people through their written names, distinguishing them from the written names of 20 unknown people chosen randomly from the phone book and assessing identification of persons recognized as familiar. In our study, names were not visually presented but read aloud by the experimenter to participants. Familiarity score was obtained by summing the number of names correctly identified as famous or nonfamous (total score: 0–60). The false alarms score (FAs) corresponded instead to the number of nonfamous names judged as familiar (total score: 0–20).

## 3. Results

Analyses were performed with the statistical software Jamovi (http://www.jamovi.org). Descripitve statistics on partipants’ scores at the experimental tests are shown in Table 2.

**(i) Unknown voice discrimination test (UVD).** An ANOVA was run to test for any significant difference in the performance of congenitally blind participants and healthy controls. The adjusted score was used as the dependent variable, while the group (two levels: congenitally blind participants; healthy controls) was insert in the model as an independent variable. Congenitally blind participants performed significantly better (M = 18.86 ± 0.98) than the healthy controls (M = 17.35 ± 1.16) [F (1,22) = 11.6, *p* < 0.005, ɳ^2^ = 0.346] (see Figure 1). We then investigated if this difference was due to the stimuli groups: As previously mentioned, UVD test stimuli are subdivided into four groups to reduce the possible interference of the sentence content. Therefore, a novel accuracy score was calculated: For each participant and for each trial, one point was assigned for each correct answer (none for wrong answers). The accuracy score was then considered as a categorial dependent variable in a linear mixed effects model where both the group (two levels: congenitally blind participants; healthy controls) and stimulus condition (four levels: same voice/same sentence; same voice/different sentence; different voice/same sentence; different voice/different sentence) were added to the model as independent variables. A by-subject random intercept was also added to account for inter-subject variability. The final model on the accuracy score (marginal R^2^ = 0.05) included a significant main effect of Group (F(1,472) = 8.04, *p* = 0.005), with congenitally blind participants performing significantly better (M = 0.95, SE = 0.01; 95% CI: 0.92–0.99) than healthy controls (M = 0.89, SE = 0.01; 95% CI: 0.85–0.92). A main effect of stimulus condition was also found (F(3,472) = 6.95, *p* < 0.001). Bonferroni posthoc comparisons revealed an overall lower accuracy for the same voice/different sentence condition (M = 0.83, SE = 0.02; 95% CI: 0.78–0.88) with respect to the other conditions (different voice/different sentence (M = 0.96, SE = 0.02; 95% CI: 0.92–1.01): t(450) = 4.01, *p* < 0.001; different voice/same sentence (M = 0.95, SE = 0.02; 95% CI: 0.91–1.00): t(450) = 3.76, *p* < 0.001; same voice/same sentence (M = 0.94, SE = 0.02; 95% CI: 0.89–0.98): t(450) = −3.59, *p* = 0.007). For the other comparisons, all ps > 0.005. Finally, no significative interaction effect (Group*Stimulus condition) was instead found (F (3,472) = 0.41, *p* = 0.740).

**(ii) Familiarity evaluation, identification, and naming of famous people from voices (VO-REC).** A series of ANOVAs were run to test for any significant difference between the two experimental groups in the familiarity, semantic, naming, and FAs scores. The adjusted scores were each time used as the dependent variable while Group was inserted in the model as an independent variable. Healthy controls made a significantly higher number of FAs (M = 4.81 ± 2.63) with respect to congenitally blind participants (M = 2.66 ± 1.94) [F (1,22) = 5.15, *p* < 0.05, ɳ^2^ = 0.19] (see Figure 1). No other significant differences were found between the two groups [Familiarity: F (1,22) = 0.09, *p* = 0.761, ɳ^2^ = 0.004; Semantics: F (1,22) = 0.00, *p* = 0.928, ɳ^2^ = 0.00; Naming: F(1,22) = 1.80, *p* = 0.194, ɳ^2^ = 0.076].

After these comparisons, we analysed sensitivity (d’) and bias (beta). As for the standard signal detection measures, Group did not differentially affect sensitivity to discriminate or the response criterion to recognize famous voices, as both d’ (F(1,22) = 0.08, *p* = 0.768, ɳ^2^ = 0.004) and criterion (F(1,22) = 2.79, *p* = 0.109, ɳ^2^ = 0.112) did not differ between congenitally blind participants and healthy controls.

**(iii) Familiarity evaluation and person identification from names (NA-REC).** A series of ANOVAs were run to test for any significant difference between groups in the familiarity and FAs scores. Raw scores were each time used as the dependent variable while Group was inserted in the model as an independent variable. No significant differences were found between congenitally blind and healthy participants [Familiarity: F (1,22) = 2.4, *p* = 0.136, ɳ^2^ = 0.098; FAs: F(1,22) = 1.31, *p* = 0.264, ɳ^2^ = 0.056].

Furthermore, Group did not differentially affect sensitivity to discriminate or the response criterion to recognize famous voices, as both d’ (F(1,22) = 0.26, *p* = 0.614, ɳ^2^ = 0.012) and the criterion (F(1,22) = 0.89, *p* = 0.355, ɳ^2^ = 0.039) did not differ between congenitally blind participants and healthy controls.

## 4. Discussion

The main purposes of the present investigation consisted of trying to confirm the hypothesis of a special link between the right temporal lobe and voice identity recognition and to understand the reasons for this special relation.

In particular, we intended to assess if the components of voice recognition that are selectively impaired in patients with right temporal lesions are specifically enhanced in subjects with an increased activation of the same strucures. More specifically, we intended to check if voice discrimination abilities and the production of FAs on famous voice recognition that are selectively impaired in patients with right temporal lesions are specifically improved in congenitally blind subjects during tasks of voice identity recognition because this could point to a common underlying mechanism. The assumption made on this subject was that the complexity of voice processing mechanisms could explain both the high number of voice discrimination errors and the high rate of FAs observed in patients with right temporal lesions. The complexity of voice processing could, indeed, hamper the formation of stable and well-structured representations necessary to evaluate if a presented voice matches or not with an already known voice. The results of our investigation were consistent with the predictions because they showed that a complete overlap exists between the components of voice identity recognition impaired in patients with right temporal lesions and those improved in congenitally blind subjects during tasks of of voice identity recognition. The only components of the voice section of the ‘Famous People Recognition Battery’ [13] on which congenitally blind subjects scored significantly better than the control sighted individuals concerned, indeed, the number of voice discrimination errors and FAs on familiarity judgement of famous voices. These were just the voice recognition tasks that have been found selectively impaired in patients with right temporal lesions by Papagno et al. [16,24]. From the theoretical point of view, this association between voice discrimination abilities and number of FAs obtained judging the familiarity of famous voices is not surprising. On familiarity judgements, perceptual (discrimination) abilities should, indeed, impact FAs more than hits because the latter require a correct matching between a percept and a stable representation, whereas the former results from a wrong matching between a percept and a nonexisting representation. More generally, it could be suggested that (a) the most difficult components of voice recognition are predominantly disrupted by brain damage and improved by brain stimulation or by an expansion of cortical areas involved in voice processing and (b) task difficulty is a factor that impacts on the degree of its lateralization. Some theoretical and empirical reasons could support these speculative assumptions. On one hand, the greater involvement of the right temporal lobe in voice than in face recognition could be due to the fact that, in familiarity assessment, the voice requires more complex processing because it relies on a temporal dimension [40,41] that could increase the difficulty of forming stable and well-structured representations, which are necessary to evaluate if a presented voice matches or not with an already known voice. On the other hand, some empirical data gathered by Brechmann and Angenstein [42] could explain the special link existing between the right temporal lobe and voice identity recognition. These authors have, indeed, studied with fMRI the impact of task difficulty on the degree of involvement of the left and right auditory cortex in the processing of complex auditory stimuli and have shown that task difficulty impacts on the lateralization of processing complex auditory stimuli. The greater difficulty of voice discrimination and voice familiarity assessment (in comparison with the analogous aspects of face recognition) could, therefore, explain the special role played by the right temporal lobe in voice identity recognition.

In the final part of this discussion, we shortly dwell on two theoretically interesting results that have been obtained in the present investigation.

The first result that could have implications for models of person recognition memory (e.g., [43]) concerns the dissociation between the better discrimination of unfamiliar voices by blind participants and no differences between the two groups at semantic levels of representation.

The second interesting theoretical aspect of the results obtained in the present investigation is that they confirmed that in a specific cognitive domain (voice recognition), a more general model of the mechanisms could subsume the greater role played by the right hemisphere in different perceptual recognition tasks.

The dissociation between the results obtained at the lowest levels (voice discrimination and FAs on the voice familiarity evaluation) and at the highest (semantic) levels of voice/person recognition is only in part surprising because it is consistent with the results of recent clinical investigations which have shown that these different levels have a diverse representation in the right and left hemisphere. The lowest levels are mainly subsumed by right temporal structures, whereas the highest levels are subsumed by the left hemisphere [44] or are not lateralized [45]. First evidence in this direction has been offered by Borghesani et al. [44] who investigated, in patients with neurodegenerative diseases, the brain regions associated with different aspects of famous face recognition disorders. Borghesani et al. [44] used the UCSF Famous Faces Battery, which comprises tasks of Famous Face Familiarity, Semantic Association, and Confrontation Naming, and showed that performance in naming and semantic association significantly correlated with grey matter volume in the left anterior temporal lobe, whereas familiarity judgment correlated with integrity of the right anterior middle temporal gyri.

Very similar results have been obtained by Piccininni et al. [45] when administering the ‘Famous People Recognition Battery’ to large groups of patients with neoplastic or degenerative damage affecting the right or left ATL. These authors observed, in fact, a greater impairment of patients with right ATL lesion at the face and voice familiarity level and worse naming scores for faces and voices in patients with left-sided lesions but no hemispheric difference at the semantic level. These results were explained by Piccininni et al. [45] who assumed that the nonverbal information provided by face and voice stimuli could be recoded (perhaps through the name of the person) into the corresponding verbal information to retrieve the critical (same profession) semantic association. Our results, therefore, support the results of these previous clinical investigations and should be taken into account by updated models of person recognition memory.

On the other hand, the observation that the expansion of the cortical areas involved in voice processing selectively improved the results obtained for tasks of voice discrimination (and on the perceptual aspects of voice recognition) is consistent with the interpretations given by De Renzi [46] and Gazzaniga [47] about the greater role played by the right hemisphere in (visual, auditory, and somato-sensory) perception. These authors suggested that this asymmetry for perceptual functions could be interpreted in terms of neural plasticity (see Gainotti [48] for a recent review) assuming that the cortical areas involved in language processing in the left hemisphere have remained dedicated to perceptual functions in homologous areas of the right hemisphere.

According to this interpretation, the mechanism of neural plasticity that, in the present investigation, had determined the improvement observed on the perceptual aspects of voice recognition should be the same that Gazzaniga [47] described with this expression: “While language emerged in the left hemisphere at the cost of pre-existing perceptual systems, the critical features of the bilaterally present perceptual system were spared in the opposite half-brain”. The only difference should be that, in that case, the left hemisphere perceptual mechanisms were decreased by the development of language, whereas in the present case, the perceptual aspects of voice recognition were improved by the expansion of the cortical areas involved in voice processing.

## 5. Conclusions

In conclusion, our study shows that a complete overlap exists between the components of voice recognition impaired in patients with right temporal damage and those improved in congenitally blind people. They suggest that the greater involvement of the right temporal lobe in voice than in face recognition might be due to the fact that in familiarity assessment voice requires a more complex perceptual processing, that could be allowed by the greater extension of cortical areas involved in perceptual functions in the right hemisphere.

## Figures and Tables

**Figure 1 brainsci-13-00431-f001:**
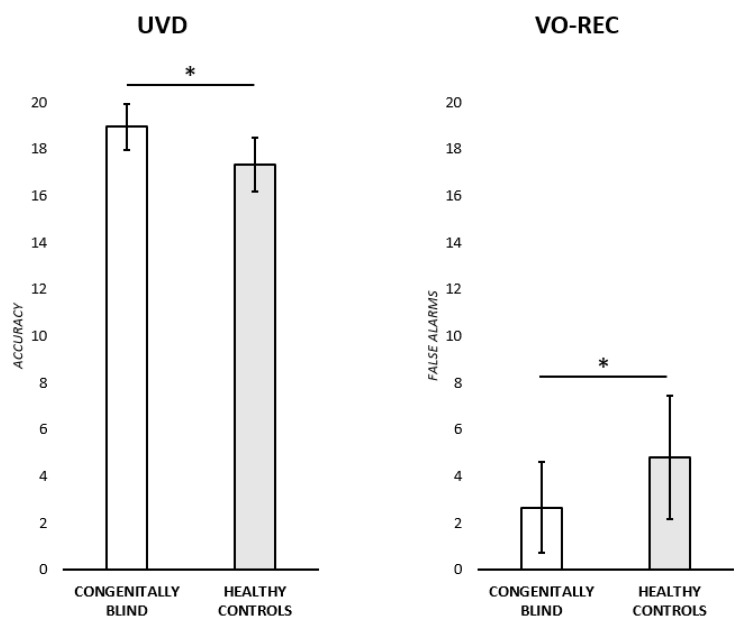
On the left, adjusted accuracy score at the unknown voice discrimination test; on the right, FAs adjusted score at the VO-REC. * Significant difference.

**Table 1 brainsci-13-00431-t001:** Demographical and clinical features of blind participants. “M” = male; “F” = female; “y” = years; “m” = month; “C” = congenital.

ID	M/F	Age (y)	Education (y)	Age of Vision Loss (m)	Residual Vision	Cause of Blindness	Age of Braille Learning (y)
1	F	39	18	C	Dark/Light	Retrolental fibroplasia	5
2	F	32	21	C	No	Agenesis of the bilateral optic nerve	5
3	F	31	18	C	Dark/Light	Microphthalmia	6
4	F	32	13	8	Dark/Light	Retinitis pigmentosa	5
5	M	31	18	C	Dark/Light	Premature retinopathy	6
6	F	33	16	C	Dark/Light	Retinitis pigmentosa	5
7	M	42	13	7	Dark/Light	Medical: retina burnt in the incubator	6
8	M	36	18	C	Dark/Light	Optic nerve hypoplasia	6
9	M	32	16	C	Dark/Light	Leber congenital amaurosis	6
10	F	35	13	C	Dark/Light	Retinopathy	6
11	F	34	18	C	Dark/Light	Premature retinopathy	5
12	F	35	13	C	Dark/Light	Premature retinopathy	6

**Table 2 brainsci-13-00431-t002:** Descripitve statistic on participants’ scores at the experimental tests. For UVD and VO-REC, adjusted scores’ mean ± SD are reported; for NA-RAC, raw scores’ mean ± SD are reported. Significant differences between the two groups are reported in bold. UVD = Unkonwn Voice Discrimination; VO-REC = Voice Recognition; NA-REC = Name Recognition; n.s. = not significant.

Test	Congenitally Blind	Healty Controls	*p*
UVD	18.86 ± 0.98	17.35 ± 1.16	<0.005
VO-REC—Familiarity	46.02 ± 5.18	45.34 ± 5.67	n.s.
VO-REC—False alarms	2.66 ± 1.94	4.81 ± 2.63	<0.05
VO-REC—Semantic	67.24 ± 19.43	67.90 ± 15.88	n.s.
VO-REC—Naming	52.11 ± 16.30	43.11 ± 16.56	n.s.
NA-REC—Familiarity	58 ± 2.76	56 ± 3.51	n.s.
NA-REC—False alarms	1.16 ± 2.28	2.5 ± 3.31	n.s.

## Data Availability

Data are available on request to the first author of this paper.

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
