# Peer review of "The Right Temporal Lobe and the Enhancement of Voice Recognition in Congenitally Blind Subjects"

_brainsci, 2023, doi:10.3390/brainsci13030431_

Round 1

Reviewer 1 Report

Review of Brain Sciences 2141604

The Right Temporal lobe and the Enhancement of Voice Recognition in Congenitally Blind Subjects

Overall:

I very much enjoyed reading this paper and I think the results are tremendously interesting.  They provide a clear demonstration of an advantage in congenitally blind participants for exactly the tasks that show a deficit in unilateral temporal lobe patients.  This cements the role of the right anterior temporal lobe when processing vocal identity, and it also indirectly suggests that the recruitment of additional cortical regions may provide a material advantage to those who have congenital blindness.

Whilst I feel that this manuscript makes a valuable addition to the questions being explored by the special issue, I have one comment which arose through my reading of the introduction and (to a lesser extent) the discussion.  This concerns the suggested comparison of voice and face processing.  In the current study, face processing is not addressed at all.  The authors note the greater difficulty of vocal identity processing compared to face identity processing (agreed) and the discussion suggests that harder tasks may show greater lateralization. As such, the authors conclude that the difficulty associated with voice processing is the reason for the lateralization of voices compared to faces.

I think that the comparison to faces is redundant when making the argument, and may even be misleading because of course face processing shows a right hemisphere lateralization as well but in occipital regions (namely the fusiform area).

With this in mind, I found the setting out of the argument a little confusing, and I might suggest that it could be streamlined to highlight (i) the lateralisation of difficult tasks, (ii) the difficulty of voice processing and thus (iii) the lateralization of voice processing before then showing (iv) the deficits for those with right temporal damage. The experimentation follows to test for benefits for those with congenital blindness. 

Comment 1 picks this point up.

Comments:

P2 line 47-51.  The findings of [17 and 18] in which a selective impairment was noted for voices than for faces when patients had damage to their right anterior temporal lobe – this difference between voice and face processing is taken to indicate greater difficulty in voice recognition than in face recognition.  Whilst I do believe that voice processing is more difficult than face processing (as in [19-23]), I am not sure that the results of [17 and 18] demonstrate this.  My understanding is that damage to temporal regions would affect voice processing, whereas damage to more occipital regions would affect face processing.  In this sense, any apparent difficulty of voice vs face processing from [18] may simply reflect the location of the brain damage in their patients.  It might be useful to revise this part of the introduction. 

P2, line 56 - typo when referring to Papagno et al. [214] (should be [24])

P2, line 62-64 – It would be useful to know whether Papagno et al.’s demonstration of a lower response criterion for voice recognition in those with a right ATL tumor was sufficient to indicate a significant bias in responding.

P2, line 89 – typo: “Drawing on this hypothesis, …”

P3, line 117 – query: You refer to haemodynamic responses in “right hemisphere and occipital areas”.   Do you mean to refer to right temporal and occipital areas?

P3, line 145 – it would be useful to know whether the test [13] uses an identical recording in the same voice/same sentence condition or whether the speakers recorded each sentence twice under different condition perhaps.

P4, line 159 – when referring to univocally identifying information – do you mean unequivocally identifying information?

P4, line 173 – In the modified naming task, you refer to the FA score as the number of non-famous voices judged as familiar but I wonder if you intended to refer to the number of non-famous names judged as familiar.

P4, line 199 – It was interesting to note the significant effect of stimulus condition in the unfamiliar discrimination task.  Can you expand further to indicate the basis for this effect, i.e., same > different, or same sentence > different sentence, or something more complex?

Reviewer 2 Report

The paper explores the voice recognition abilities of blind individuals compared to sighted individuals which is a fascinating topic and fairly understudied. The introduction reviews literature in patients with right temporal tumours and discusses the brain regions involved in voice processing tasks in blind individuals. Unfortunately, the lack of explicit hypotheses to be tested, and how the authors will test them with their chosen paradigm and measures, detracts from the theoretical importance of the work presented. My main issue is that you have these different voice recognition measures, but what is the rationale for this? What hypothesis was it testing? This needs to be explained in the Introduction. The Methods could do with some serious rewriting and the authors should use signal detection methods to analyse their results. Signal detection analyses would allow for more conclusive interpretation of the results and would offer a more appropriate means to compare the findings with previous literature on voice recognition measures in sighted and blind individuals. As it stands, it is difficult to rule out any differences in response criteria between the two groups based on the analysis of FA alone. I think the overall topic is interesting and worthy of investigation and the (potential) central result, that blind individuals are better at unfamiliar voice discrimination than sighted individuals, but no group differences at semantic levels of representations. However, I am concerned that the writing, analyses and presentation of the results fails to sufficiently communicate the findings. I think some rewriting and restructuring is needed. 

Page 3, l.23. Could the authors please clarify what they mean by ‘opposite conditions’ in the following statement:

“However, to better evaluate the relations between the impairment of voice recognition documented in patients with right temporal lesions and the improvement observed in congenitally blind individuals, it could be interesting to judge if the same components of voice comprehension are involved in these opposite conditions.”

Introduction and Discussion would benefit from the inclusion and discussion of the following literature:

Recognition famous voices: Influence of stimulus duration and different types of retrieval cues.

Schweinberger, Stefan R;Herholz, Anja;Sommer, Werner. Journal of Speech, Language, and Hearing Research; Apr 1997; 40, 453-463.

The voice-recognition accuracy of blind listeners. Perception, 12, 223 - 226.

Bull, Rathborn & Clifford (1983).

Report on blind subjects’ tactile and auditory recognition for environmental stimuli. Perception and Motor Skills, 48, 363 - 366.

Cobb, Lawrence and Nelson (1979).

Page 3, l. 120-128. Introduction provides the reader with a rationale for performing the research, but is lacking in clearly presented hypotheses. Please could the authors provide the hypotheses for the study. 

Materials and Methods

Page 3, l. 130. Could the authors clarify whether all blind participants were totally blind or did not have more than rudimentary sensitivity for brightness differences without any pattern vision?

Page 3, l.141 – 147. i) Unknown voice discrimination test (UVD). 

It is not clear how the total score of 20 points is allocated across the four groups: 1) same voice/same sentence; 2) same 145 voice/different sentence; 3) different voice/same sentence; 4) different voice/different sentence. Could the authors please clarify? If the participants’ task was to judge if the voices belonged to the same or to different people what would a sample trial look like for each group? For example, if the first audio file was selected from group 1 (same voice/same sentence) what would the second audio file consist of? What would be classed as a correct response and an incorrect response?

The authors state that the two audio files lasting about 15 seconds were consecutively presented? Was there any kind of delay between the two audio files? Please specify the gap between the two audio files, if any. What was the content of each sentence? Did it differ in each audio file?

Pages 3-4, l. 148 – 162. ii) Familiarity judgment, recognition, and naming of famous people from voices (VO-REC) 

More information required regarding the selection of stimuli. The authors should provide the names of the famous persons and the domains that they were selected from in an Appendix (please refer to Damjanovic & Hanley, 2007; Schweinberger et al., 1997). The authors should also outline the procedures and analyses performed that validates their assertion that each audio fragment “did not contain any element that could allow the direct recognition of the person”, as this has often been an overlooked confound in previous research (see Damjanovic, 2011; Hanley & Damjanovic, 2009).Please also consult the following references:

Barsics, C., & Brédart, S. (2011). Recalling episodic information about personally known faces and voices. Consciousness and Cognition, 20, 303–308. 

Brédart, S., Barsics, C., & Hanley, J. R. (2009). Recalling semantic information about personally known faces and voices. European Journal of Cognitive Psychology, 21, 1013–1021.

Damjanovic, L. (2011). The face advantage in recalling episodic information: Implications for modeling human memory. Consciousness and Cognition, 20, 309-311.

Damjanovic, L., & Hanley, J. R. (2007). Recalling episodic and semantic information about famous faces and voices. Memory and Cognition, 35, 1205–1210. 

Hanley, J. R., & Damjanovic, L. (2009). It is more difficult to retrieve a familiar person’s name and occupation from their voice than from their blurred face.  Memory, 17, 830–839.

Hanley, J. R., Smith, S. T., & Hadfield, J. (1998). I recognize you but I can’t place you: An investigation of familiar-only experiences during tests of voice and  face recognition. The Quarterly Journal of Experimental Psychology, 51A, 179–195.

Hanley, J. R., & Turner, J. M. (2000). Why are familiar-only experiences more frequent for voices than for faces? Quarterly Journal of Experimental Psychology, 53A, 1105–1116.

More information is required regarding how the semantic score was calculated. The authors state that “two multiple choice questions investigating the general and specific categories to which the famous person belongs” were used (page 4, l.157-158). Please provide the number of alternative responses in the multiple choice questions, their presentation format (i.e. ,read aloud by experimenter?), state where the correct answer was presented in the selection (i.e., randomised for each question?) and describe the process in selecting the foils.  How did the authors assess and score the “univocally identifying information about the person” provided by the participants (page 4, l.159). Please consult the following references:

Gollan, T. H., & Brown, A. S. (2006). From tip-of-the- tongue (TOT) data to theoretical implications in two steps: When more TOTs means better retrieval. Journal of Experimental Psychology: General, 135, 462-483. 

Hanley, J. R., & Damjanovic, L. (2009). It is more difficult to retrieve a familiar person’s name and occupation from their voice than from their blurred face.  Memory, 17, 830–839.

Page 4, l. 163 -174. iii) Familiarity evaluation and person identification from personal name 

Could the authors provide further information about the unknown people chosen randomly from the phone book - how were the names matched for word length and number of syllables to the names of 40 Italian celebrities?

3. Results 

Page 4, l.179-180 The authors state that “for the UVD test, the adjusted score was used as dependent variable.” Could the authors clarify what is the adjusted score. The authors should analyse their data with signal detection methods by incorporating both hits and false alarms to establish differences (if any) in sensitivity and criteria between the two groups. This type of analysis would enable better comparisons to be made with the previous literature. 

Page 4, l.80 – 182 The authors also state that “for the VO-REC, familiarity, FAs, semantic and naming adjusted scores were each time used as dependent variable.” Please explain how the adjusted scores were calculated and please include statistical analyses that compares the semantic scores to chance levels based on the multiple choice questions.

Page 4, l. 183-184. When the authors describe their results in terms of “familiarity and FA raw score” are they referring to “familiarity” as a “hit”. If so, please re-name as ‘hit’ throughout and apply signal detection methods (consult recommended literature previously provided).

i) Unknown voice discrimination test (UVD). 

The dependent variable is really not clear here at all. Please simplify and clarify. What is a correct response? What is a false alarm in the UVD? If the data can be coded as proportion hits and false alarms, signal detection analyses should be conducted for measures of sensitivity and criteria. Treating accuracy as a categorical variable without separately considering error rates is confusing. Signal detection analyses seem more appropriate for the purpose of this study and should then be entered in an independent groups comparison (with 2 levels: sighted controls or blind participants). It is still a little unclear to me what the stimulus condition actually is and how this translates into performance accuracy and error. As such, I’m not entirely convinced what the additional stimulus condition by 4 levels adds to the analyses. Suitable effect sizes need to be reported alongside all inferential tests.

ii) Familiarity evaluation, identification, and naming of famous people from voices (VO-REC). 

Page 4, l.202 – 206. Descriptive statistics should accompany the inferential statistics reported for these measures: No other significant differences were found [Familiarity: F (1,22) = 5.15, 205 p = .761; Semantics: F (1,22) = 5.15, p = .928; Naming: F(1,22) = 5.15, p = .194]. In the absence of a more detailed explanation how the semantic score was calculated (see previous comment), it is difficult to interpret these results in the context of previous research. Suitable effect sizes need to be reported alongside all inferential tests.

iii) Familiarity evaluation and person identification from names (NO-REC). 

Page 4, l.207 – 209. Again, descriptive statistics should accompany the inferential statistics reported for these measures: [Familiarity: F (1,22) = 2.4, p = .136; FAs: F(1,22) = 1.31, p = .264]. Results should be reanalysed using signal detection methods. Suitable effect sizes need to be reported alongside all inferential tests.

Page 5. Figure 1 is not clearly presented. The rule for the vertical axis is absent and the font size is too small. It’s not clear why the right panel combines the FA data for the Familiarity evaluation, identification, and naming of famous people from voices. Responses to the semantic categories should be presented separately.

Discussion

Page 5, l.214 -216. This section begins by stating  that “The main purposes of the present investigation consisted in trying to confirm the hypothesis of a special link between right temporal lobe and voice identity recognition and to understand the reasons of this special relation.”, but no clear exploratory or directional hypotheses are presented at the end of the Introduction section. The authors should make a much clearer effort to present their hypotheses in the Introduction in order to improve the theoretical importance of the results interpreted in the Discussion section.

It would be helpful to see a more detailed discussion on how performance with the semantic responses compares to previous research (see recommended reading provided). In terms of gaining a deeper theoretical appreciation of the study’s results, what are the implications for models of person recognition memory (e.g., Bruce & Young, 1986)? How can the authors explain better discrimination of unfamiliar voices by blind participants, but no differences between the two groups at semantic levels of representation. This is an important consideration to be made and should be elaborated on in more detail to highlight the theoretical significance of the work reported and some of the arguments made in the Discussion, page 5, l.242 - 245: “On familiarity judgements, perceptual (discrimination) abilities should, indeed, impact more on FA than on hits, because the latter require a correct matching between a percept and a stable representation, whereas the former result by a wrong matching between a percept and a non-existing representation.” 

Bruce, V., & Young, A. W. (1986). Understanding face recognition. British Journal of Psychology, 77, 305–327.

Round 2

Reviewer 2 Report

The authors have made some effort to address some of my concerns, but a number of fundamental issues still remain. Specifically, the Introduction section is lacking a clear set of hypotheses for the study, further statistical tests for chance levels have not been performed for general and specific 4AFC responses. Attention to detail is also lacking, with some effect sizes reported and others omitted. Overall, these issues would need to be explicitly addressed (as requested in my first review).
